# The Best Welfare Deal: Retirement Migrants as Welfare Maximizers

Inés Calzada *, Virginia Páez, Rafael Martínez-Cassinello and Andrea Hervás

Faculty of Political Sciences and Sociology, Complutense University Madrid, 28040 Madrid, Spain;
rafmar07@ucm.es (R.M.-C.); andrea.hervas94@gmail.com (A.H.)
* Correspondence: icalzada@ucm.es

**Abstract:** Retirement migration within Europe has increased enormously since the 1990s. It now involves millions of elderly Europeans moving from Central/Northern Europe (UK, Germany, Scandinavia) to Mediterranean countries (Malta, Portugal, Spain) in search of a better quality of life. Most previous research departs from an ethnographic perspective to look at the personal experiences and motivations of retirement migrants. In this paper, we adopt a macro-level perspective to address the use that retirement migrants make of the European framework of social rights. We aim to understand (a) to what extent do retirement migrants living in Spain ask for help from the local Social Services when they enter into dependency? (b) Do retirement migrants engage in strategies to maximize their welfare rights? To answer these questions, we carried out qualitative phone interviews with the coordinators of Social Services that cover 80 (out of 119) of the Spanish municipalities with larger numbers of retirement migrants (more than 30% of elderly residents are foreigners).

**Keywords:** retirement migration; Social Services; dependency; social work; elderly care; mobility; intra-European migration





## 1. Introduction

Retirement migration is defined as the movement of people after working-age for reasons of leisure and lifestyle. As a difference from other types of migration in old-age, retirement migrants are not mainly driven by economic reasons nor family reunification but for climate, landscape, lifestyle, and social life (although changes in pension systems can increase the importance of economic drivers). After many years of increase, retirement migration is nowadays one of the major migration trends among elderly Europeans moving within Europe.

The impact of international retirement migration on the origin and destination places is difficult to quantify because many of the elderly migrants never register as residents in the destination places. Since the beginning of the phenomenon, authors have worried about the impact of this growing flux of people on national Welfare States [1]. Particularly because retirement migrants move from well-funded Welfare States (Northern and Continental Europe) to their less developed neighbors (Spain, Portugal, Greece, Malta), and as European citizens, they are entitled to receive social protection in any EU country where they wish to live.

International Retirement Migrants (IRMs) are a very interesting case for studying the complex relationship between mobility and welfare rights. International Retirement Migrants (IRMs) move, for the most part, thanks to the portability of old-age pensions. In this respect, they are a case of success for the idea of a 'Social Europe' and the EU Framework for Social Protection. However, many IRMs require some care services if they want to remain in the host country, and the little information that we have does not suggest a good fit between EU legislation and their care needs. In the next pages, we will try to understand, even if partially, the uneasy relationship between public social provision and retirement migrants using Spain as a case study.

This paper presents the first results of a research project on the relationship between retirement migrants and public Social Services in Spain (funded by the Spanish Ministry of Science) that aims to: (a) get to know the extent to which European retirees living in Spain make use of public Social Services and benefits; (b) identify the circumstances in which they come into contact with the Social Services; (c) understand the challenges that these new users pose to social workers; (d) identify and evaluate the strategies developed by said workers to assist them.

To fulfill these aims, we used a mixed methods design including phone interviews with the coordinators of local Social Services in 119 municipalities (i.e., all Spanish municipalities where more than 30% of elderly residents have a non-Spanish nationality); in-depth case studies in five destination places, and an online survey distributed to social workers in destination areas.

For this paper, we have created a database with information about the 119 municipalities with a high presence of EU retirees. In this database, we have merged information about the use of Social Services by IRMs obtained through phone interviews, Local Registry data about the nationality of the EU registered residents, and economic information about the average income in each municipality. We show, for the first time, an exhaustive quantitative panorama of the relationship between retirement migrants and public Social Services across all Spanish destination areas.

## 2. Theoretical Framework

### 2.1. Retirement Migration and the Welfare State

International retirement migration (IRM) is a particular type of lifestyle migration that certain individuals engage in after their retirement from regular work. At a difference from other forms of later-life migration, retirement migrants move voluntarily to places characterized by a better climate and a wider range of amenities [2].

Almost 20 years ago, Warnes and colleagues stated that "the number of older migrants in Europe (and other developed world regions) will grow substantially during the coming half-century" [3] (p. 308). The forecast has come true, as international retirement migration (IRM) has increased enormously since the 1990s and now involves millions of elderly Europeans. Figure 1 shows the evolution of registered residents in Spain, over 65 years old, coming from five key origin countries. In less than twenty years, elderly residents from the Netherlands, France, UK, Belgium, and Germany increased by 300%, reaching more than 200,000 persons in 2020. Recall that all previous studies indicate that many EU elderly living in Spain never register as residents, and that registration has to be multiplied by 3 or even by 5 to get a glimpse of the real numbers [4,5]. Duran finds evidence that the British community tends to register more than the German, but we do not have a clear picture of which nationalities are more or less underrepresented in local population registers [6].

With data from the Spanish National Population Register, on the 1 January 2021, there were 267,625 European citizens (EU 27 + UK) older than 65 years old registered as residents in Spain. Of them, 28,491 were Germans and 100,123 were citizens of the UK. Taking into account the fact that many people arrive in Spain before turning 65 years old (due to early retirement or different national policies concerning old-age pensions) and the extended under-registration among this type of migrants, it is not an exaggeration to say that at least one million elderly Europeans could now be living in Spain.

International Retirement Migration (IRM) has been extensively studied. However, most of the literature on this phenomenon deals with the experiences and daily routines of retirement migrants in the host country whilst they are autonomous. Studies detail the socio-demographic characteristics and motives behind the migration of IRMs [5,7,8], the daily routines and activities of retired migrants [9,10], the problems encountered in the process of settlement [11], the strategies developed to keep in contact with friends and family in the home country [12], their lifestyle choices inherent to their decision to migrate [2], the pursuit of self-realization, self-exploration and self-development [13], their housing preferences [14], their charitable and volunteering activities [15], and so on.

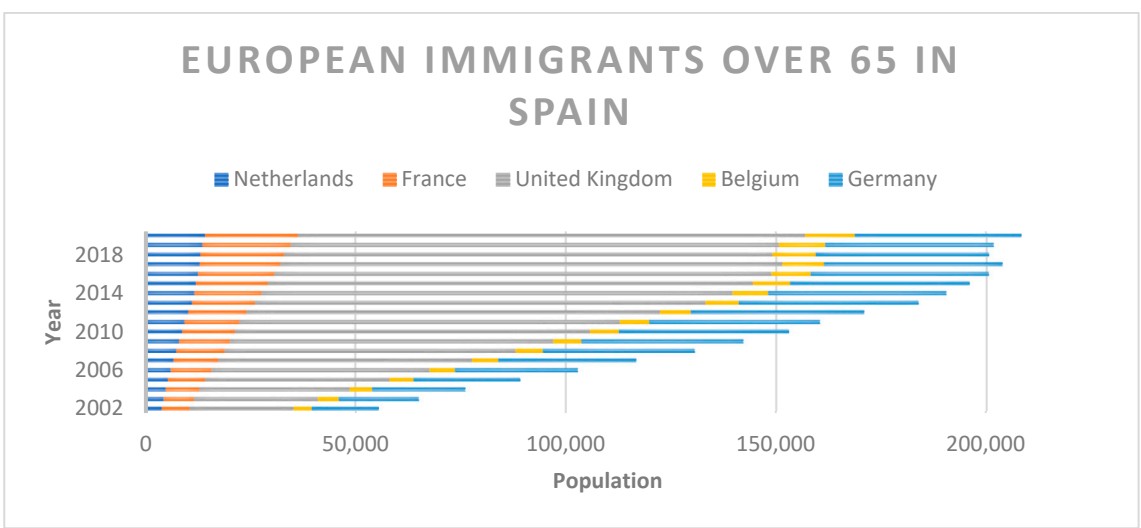

**Figure 1.** People over 65 years from five EU countries registered as residents in Spain. Source: Own elaboration with data from the Spanish National Statistical Institute (INE).

This focus on the self-sufficient 'active elderly' leaves somewhat in the shadows two issues that are important for this paper: First, the fact that retirement migration involves other actors apart from the elderly migrants: people from the same origin countries than retirement migrants have moved to the same destination areas to make a living working for these mobile and (part of them) well-off retirees. In areas with high density of retirement migrants, a burgeoning market of services oriented to this group has developed, offering food, leisure, restoration, and medical and care services in their own language and according to their own social norms for those who can afford it [16,17].

Secondly, the time factor that, in this particular case, can generate drastic changes in the situation of elderly migrants in the short-run is another issue. What happens when retirement migrants lose autonomy? Do they return in order to benefit from the Social Services that they know? Or do they stay and use public care services in the host country (since they are entitled to)? Can they pay for the cost of private care provision in the host country?

For many years, there was the assumption that the vast majority of retirement migrants chose to return to their native countries when they started losing autonomy [18–21]. However, recent research indicates that substantial numbers do stay in the destination areas well after dependency sets in, using complex strategies that involve public, private, and voluntary/social resources [15–17,22]. Ethnographic studies have also shown the complexity of this decision to return after 15 or 20 years living abroad. For many retirees, this move 'back home' means a return to a country where they have no home, no social connections nor routines—a return that many would rather avoid.

However, to stay in the host country until the end of life is not exempt from significant risks derived from a lack of preparations and anticipation of migrant retirees [18–23]; language and cultural barriers [15,22,24,25] or having insufficient financial resources to get an adequate level of care services in the destination country [17].

*2.2. The Welfare State for Mobile Retirees*

The concept of a 'Social Europe', put forward by Jaques Delors, has been subject to debate since its birth. However, in its basic form, this idea is just a reflection of a common pattern across EU countries regarding the social rights granted to the population. Notwithstanding its diversity, all EU countries have put into place organized systems of social protection including healthcare, education, and public pensions, making the EU one of the areas with a higher level of public social expenditure [26].

Apart from sharing the common goal of protecting citizens from social risks, social protection systems differ among countries in almost every respect: in the size of public budgets, in the welfare areas under public supervision, in the organization of public programs, in the eligibility criteria to access benefits, etc. A vast amount of literature on comparative welfare politics has aimed to find patterns in this variation [27,28]. There is now an agreement on the existence of (at least) three groups of countries regarding the similarity of their welfare arrangements: Social-democratic: with a wide layer of programs that provide generous benefits for all residents on an universal basis (healthcare, minimum income, child and elderly care, family benefits), and also contributory programs for pensions and unemployment (Nordic countries); Corporatist: based mostly on contributory programs in which your benefits depend on your contributions (amount and duration) and where it is usual to have differences in provision according to occupational groups (France and Germany); Liberal: health-care and education are universal but many programs are means tested (grant benefits only for low-income people). Public pensions are low and people are expected to combine public protection with private protection (pensions) and family care (UK, Switzerland).

Notwithstanding the differences, efforts have been made so that EU citizens can move not only between countries but also between welfare systems without losing the safety network that any Welfare State is meant to guarantee. Thanks to the EU Framework of Social Rights, some of the welfare benefits can be transferred from one country to another, particularly old-age pensions. Concerning Social Services and benefits, EU citizens can claim social benefits in another EU country after having been registered as a resident for five consecutive years. It seems quite easy. Retirement migrants can bring their old-age pensions with them and, when in need of healthcare or social assistance, benefit from the NHS and Social Services of the host country. The reality is somewhat more complex.

Spain, as other Mediterranean countries, fits poorly into the welfare typologies. Italy and Spain, for example, combine programs similar in design to those in the Nordic countries (NHS; contributory pensions) with others that resemble those of Liberal Welfare States (Social Services, minimum income). This has led various scholars to suggest the need to include a fourth type of Welfare State: the 'Southern', 'Mediterranean', or 'Familistic' welfare model [29–34]. This last term is possibly the one that best captures the idiosyncrasies of the model, since the entire system is based on the assumption that although the state is responsible for providing healthcare, education, and pensions, all care needs should be provided for within the family.

The ranking of countries in terms of traditional family forms proposed by Kohli et al. [35] accords with this organization of the WS, "with the Scandinavian countries generally having the least traditional family structure, the Mediterranean countries (Spain and Italy more so than Greece) the most traditional one, and the Continental countries lying somewhere in-between". Mediterranean countries have the highest levels of intergenerational cohabitation and spatial proximity of family members; care activities carried out by family members are more time-intensive in the South than in other countries [36], and adult children provide personal care to their elders (dressing, bathing, eating) much more frequently than in the Nordic or Continental countries [37–39].

This reliance on the family to provide care for dependents (children and the elderly) is of enormous importance for those retirement migrants that decide to stay in Spain (or in any other Mediterranean country) even after losing some autonomy. The vast majority of retirement migrants move only with the spouse/partner, have no family connections in the destination country, and may be alone after some years if the spouse/partner dies.

If the retiree decides to stay in the destination country and needs some care services she/he will be faced with a welfare system that speaks a different language, is organised according to different cultural norms, has particular criteria to access benefits, and assumes the existence of family members to complement public support.

Hall and Hardill [18] studied the elderly British in Spain and described their difficulties accessing the Spanish NHS. These difficulties are related to lack of language proficiency by

IRMs: lack of enough translators in the Spanish NHS, and absence of the family networks that the Spanish care system expects to step in when the person leaves hospital or even during her/his stage in it. Results on the use that IRM make of Spanish NHS are heterogeneous: Casado-Díaz [7] implemented a non-random survey of elderly retirees who own houses in Torrevieja (Spain), finding that a large majority 'rarely' or 'never' used the Spanish public healthcare services. By contrast, the ethnographic research conducted by Legido-Quigley and McKee [40] on British pensioners living on the Spanish coast concluded that these individuals extensively use (and are satisfied with) these services.

Regarding public Social Services, Hall and Hardill [18] and Calzada [24] identified an under-use of care services by retirement migrants, a phenomenon derived from a combination of misinformation; communication problems; non-compliance with basic eligibility criteria (such as being a registered resident for more than five years, having low income, etc.); distrust of the Spanish system, and a quest for autonomy. Taking into account this overall description, it is easy to see that retirement migrants move between very different welfare systems, and the systems differ particularly in what retirees may need more: care help for dependency and Social Services for the elderly.

### 2.3. Social Services Available for Retirement Migrants in Spain

Up until 2006, Spanish state intervention in care consisted of a means-tested, underfunded, fragmented, and decentralized scheme of personal services run by regional governments and municipalities that catered only to those in the most urgent need. Retirement migrants had little chance to access these benefits because, even if they would comply with the requirement of being registered residents in Spain for more than five years, their old-age pensions were over the threshold to access benefits.

In 2007, the country witnessed a major change with the creation of a new national public program called SAAD (System of Autonomy and Dependency Care). This program offers public help (in the form of cash transfers or direct services) for all residents in Spain with any level of dependency. Although co-payments according to income are in place, SAAD is universal and aims to achieve a full coverage of the care needs of dependent people that have been registered as residents in Spain for the previous five years before claiming benefits.

However, there are complications: to be granted public help, one needs to undergo a complex bureaucratic process and depending on the region, it may take more than two years to get the benefits or services. These delays explain that, at present, the SAAD coexists with the old system of means-tested benefits, which steps in for those that have been unable to achieve care services under the SAAD.

The SAAD is universal, offers care services for any level of dependency, and does not require low income to receive public help. With these conditions, it should be of interest for retirement migrants and, actually, there are worries about the financial capacity of the Spanish SAAD to provide care services not only for Spaniards, but also for the EU elderly residents.

In the next pages, we will evaluate novel empirical data on the relationship between retirement migrants and the local Social Services in an attempt to see if this group of mobile citizens is characterized by a strategic approach to social rights, i.e., we want to see if patterns of registration/non-registration as residents in Spain and the subsequent use of Spanish Social Services are coherent with strategies to maximize the 'welfare package' which the different nationalities can obtain by cleverly navigating across welfare systems.

### 2.4. Hypotheses

Our first hypothesis concerning these patterns is as follows:

**Hypothesis 1.** *Levels of registration in Spain by EU elderly correlate with levels of claims to the local Social Services by this population. Registration would then be employed as a means to use the destination country's welfare system.*

According to EU law, healthcare and social protection for Europeans living in a different EU country than their own fall almost entirely on host countries. In the case of healthcare, EHIC (European Health Insurance Card) covers medical assistance for a brief period of time, and migrants are expected to be insured in their country of destination, whether privately or accessing public provision through legal registration in the country. Regarding Social Services such as income support or help for dependency, home countries follow the logic that assistance will not be given if it would normally be expected from the country of residence. Thus, to be able to enjoy and benefit from social rights in the host country, it is imperative for retirement migrants to register in the new country of residence. We expect to find a correlation between levels of registration of EU elderly in a municipality and levels of claims to the local Social Services by this same group because, although registration in Spain is compulsory, there is no supervision of the rule and Europeans are never asked to show registration documents. We know that substantial numbers of retirement migrants never register as Spanish residents even after many years of living in the country. In the absence of inspections or any form of control by Spanish authorities, registration is just an option. If retirement migrants act as welfare maximizers, registration shall be used only, or mostly, when one wants to access public healthcare and social assistance in Spain.

Of course, each country has its own national policy concerning social rights for its diaspora, which can generate different care strategies for different nationalities. However, national policies in this regard are similar in the Nordic countries, the UK, and Germany (which constitute some of the largest groups of retirement migrants in Spain). Whether it is Germany, Sweden, or the UK, social rights are based on residence and nationals living abroad receive little attention in political debates about welfare benefits and have a very limited (or no) access to Social Services [41–43]. Governments' engagement with their nationals living abroad is characterized for being limited and modest in the field of social protection. There is no targeted assistance or support intended specifically for this population, and this social protection is only available to those living abroad on the basis of exception (emergency and unusual circumstances). Klekowski von Koppenfels [41] comments on the German case saying that:

> "*Germany seems not to think of Germans abroad living long-term or permanently abroad as a population of citizens maintaining ties, or needing assistance, but rather as having become citizens or long-term residents elsewhere, perhaps because of its long history as an emigration country. Indeed, there is neither an explicit emphasis on German emigrants nor are there many policies in place which are intended specifically for them. Rights are largely granted on the basis of residence (in Germany), rather than (German) citizenship. Two exceptions do emerge, with access to voting rights and to pension rights strongly facilitated. Other rights are granted only on an exceptional basis, with consular authorities playing a role in facilitating that access*".

In the same vein, the UK focuses on providing comprehensive online information for Britons living abroad (GOV.UK) [44], but has a complete absence of policies to facilitate their social protection abroad, driven by the assumption that this population do not want or need to engage with their homeland state. In the case of Sweden, the country lacks a central policy regarding its emigrant population, and therefore, the legal framework is often confusing. Swedes living in Spain frequently seek support in less formalized ways via the Swedish Church or other organizations.

The major exception to this aforementioned rule concerns contributory old-age pensions, which are portable and accessible to those moving abroad. Actually, it is the portability of pensions which allows the very existence of retirement migration because, apart from some exceptions, the vast majority of elderly Europeans living in Spain have their old-age pension as their main income.

So, if registration is necessary to claim social protection in the host country and it is also compulsory: Why do many EU retirees living in Spain officially figure as residents in their home country despite potential fines? Moreover, why do others prefer to register?

Registration in Spain means de-registration in the home country and, correspondingly, losing the right to access social protection there. If retirement migrants are welfare maximizers, the answer shall lie in the welfare trade-off of this decision. Some Welfare States are more generous than others, and hence, de-registering from these is more 'costly' in terms of welfare rights. Other WSs are more meager, and de-registering from them to access another meager WS may not be very detrimental. Having in mind the welfare typologies and comparative studies on social protection across the EU [27–30], our second hypothesis is that UK citizens will have more incentives to register than German citizens. Of course, we cannot know what share of Germans and British that live in Spain do actually register because there is no way of measuring un-registered residents. Hence, we have two hypotheses that address this difference in registration indirectly:

**Hypothesis 2.** *UK elderly citizens are more present among users of local Social Services than German elderly citizens. Municipalities with a high share of UK citizens among their residents are characterized by a high presence of UK seniors among claimants of Social Services.*

**Hypothesis 3.** *UK citizens are more present among users of local Social Services even in municipalities that are key destination spots for Germans.*

Our second and third hypotheses assume that retirement migrants living in Spain will be more or less prone to register as residents in Spain depending on the generosity of the Social Services in their home country. Nationals of countries whose Social Services are similar to those in Spain will tend to register and, when in need, claim social assistance in Spain. Nationals of countries where Social Services cover a wider range of needs and/or are more generous than their Spanish counterparts will tend to avoid registration to keep social rights in their home country.

The hypotheses above are based on the fact that, coming from a not generous Welfare State (WS), British people will act strategically and register in Spain to be able to use the Spanish WS and, as a bonus, comply with the rules.

## 3. Materials and Methods

### 3.1. Research Project

Results presented in this article are part of a wider research endeavor funded by the Spanish Ministry of Science and Technology under the title "ANOMYM". This project defines its objectives on two levels: at the micro level, we seek to understand the relationship between European retirees living in Spain and the Social Services; at the macro level, we want to map and explain geographical variations in this relationship.

The first objective involves getting to know: (a) the extent to which European retirees make use of the Spanish Social Services and benefits; (b) the specific cases in which they come into contact with the Social Services net (dependency, lack of resources, economic crisis, return assistance); (c) the challenges social workers face posed by these new users, and (d) the strategies developed by said workers to assist them. This knowledge can be helpful in designing new social policies at all levels, but mainly at the municipal level.

From a previous preliminary study, we know that, even comparing municipalities with a large number of migrant retirees, these residents approach public Social Services to varying extents depending on the municipality. There are places with huge numbers of elderly European residents where social workers never encounter them as claimants or users of the Social Services; whilst in other municipalities with similar numbers of retirement migrants, social workers are stressed and overburdened by the demands posed by this group of residents. In that preliminary study, we could not find any convincing explanation for this variation (ANONYM). The present project, that involves much more extensive fieldwork, aims to put forward novel information that helps to understand why retirement migrants are much more prone to claim their rights to social assistance in certain areas.

*3.2. European Retirement Places*

Our study stands out from most previous research on retirement migration for its macro-level perspective. As we explained in previous sections, most studies about retirement migrants adopt an ethnographic perspective. This is partly due to an interest in the lives and motivations of the mobile elderly and partly due to sheer difficulties in obtaining reliable, quantitative information on the actual number of retirement migrants living in settlement areas. Although we can use data from Local Population Registers, ethnographic studies repeatedly find that many retirement migrants never register as residents in their municipalities of settlement [4,5,15,45–47]. Under-registration can generate a problem of under-funding for Social Services in municipalities that attract many retirement migrants, since public funds for municipalities are calculated based on the registered population, and these areas have many more residents than those actually registered [18,48,49].

The impact of retirement migration on the destination places will be important, particularly because European retirees come from all over Central and Northern Europe, but they settle into very particular places. This spatial concentration is such that, already in 2004, a group of scholars working on retirement migration created the term "European Retirement Places" ("Lugar Europeo de Retiro"), defined as:

*"European Retirement Places (LER) are those in which, as a joint effect of the progressive aging of the population; the increase in the standard of living and autonomy of citizens; as well as the possibilities opened up by European integration, a large contingent of elderly citizens (the vast majority retired) from other Member States of the European Union are settling. Those elderly citizens have a vocation of residing in the new places more or less permanently"*. [50]

In this project we have 'operationalized' this idea of European Retirement Places as municipalities where at least 30% of the registered residents older than 65 years old have a non-Spanish citizenship. As we explain subsequently, of the 8131 Spanish municipalities, we found 119 that comply with this criterion.

*3.3. Data Collection and Methods*

The methodology of this project combined qualitative and quantitative methods to gather primary data.

The qualitative approximation involved (a) short qualitative interviews with Social Services' coordinators in 119 municipalities identified as European Retirement Places. Interviews were carried out by phone and lasted between 15 and 25 min. Notwithstanding all our efforts, we managed to interview the coordinators of Social Services in only 80 of these 119 municipalities; (b) ethnographic studies of five European Retirement Places. Each case study involved observation, collection of secondary data and printed material (leaflets, local newspapers, etc.), long personal interviews with different actors related to retirement migration in the village, and discussion groups.

The quantitative approximation included (a) an online survey distributed among social workers based in any of the 119 European Retirement Places and working in public, private, or voluntary premises (non-representative, snowball sampling); (b) the creation of a database with information from our 119 European Retirement Places. This database merges information from official sources with information derived from the telephonic interviews with coordinators of Social Services.

The mix of methods aims to collect different types of information on the same phenomenon (nuanced in the ethnographic case studies; simple but representative in the short telephonic interviews) and to ground the validity of the conclusions.

In the next pages, we present information derived mostly from the telephonic interviews with coordinators of Social Services and from the analysis of the database mentioned above. These telephonic interviews had questions that aimed to quantify the impact that retirement migration has on Social Services (such as the share of retirement migrants among

users of Social Services during the previous years or their most frequent demands), and also general questions on social workers' perceptions about retirement migrants.

## 4. Results

### 4.1. General Panorama of European Retirement Places

Using local population registers of the whole country, we identified all Spanish municipalities where more than 30% of the residents over 65 years of age had a non-Spanish passport (data from the Spanish Statistical Office—INE). The limit of 30% was set because a previous study by Rafael Durán detected the existence of "some municipalities where more than one third of all residents over 65 years of age hold a European nationality other than the Spanish" [8].

Working with Register data for the whole country, we computed, for each municipality, the percentage of the registered elderly (over 65 years old) that had a non-Spanish nationality (all foreigners, not just Europeans). One may believe that this would increase the number and types of places found, but this was not the case. In all municipalities where there is a high number of elderly foreigners, those foreigners are Europeans. All the municipalities where elderly foreigners amount to a substantial share of the elderly population are on the coast and accord to the spatial preferences of European retirees (warm climate, close to the sea, touristic infrastructure, etc.).

In total, we found 119 municipalities where at least 30% of the elderly were foreigners. Most of the municipalities are located in three regions and, within these regions, in a handful of provinces: Málaga and Almeria, in the region of Andalucía; Alicante, in the region of Valencia, and the two largest islands of the Canary Islands: Tenerife and Gran Canarias.

This distribution shows two important things:

1.  Retirement migration is a spatially concentrated phenomenon and European Retirement Places are quite stable along the years. The provinces where our municipalities are located are known for being IRM destinations since the 1990s.
2.  The presence of non-Spanish elderly people is unusual in other areas of the country. Although 10% of residents in Spain have a foreign nationality, the percentage falls to 3% among those over 65 years old. The province of Madrid (where the Spanish capital is located) features 13% foreign residents, but only 2% of residents over the age of 65 have a foreign nationality.
3.  An additional interesting result that one can derive only by looking at the list of European Retirement Places is that in many places, the share of Europeans among the elderly is well above 30%. Considering that our data are, for sure, under-estimating the real number of elderly foreigners, our results point to the existence of many Spanish municipalities where retirement migrants are a very relevant group in size.

Table 1 serves as a resume of the elderly population in European Retirement Places. In 60% of the 119 municipalities, elderly foreigners are more than 40% of the elderly population. This increases our expectation that this group of residents shall have an impact on the life of the village and, particularly, in the public services for the elderly that are offered by local councils.

**Table 1.** Elderly foreigners as a share of the elderly population.

| RESIDENTS AGED 65+ | | N Municipalites | Percentage |
|---|---|---|---|
| | 30–40% | 48 | 40.3 |
| **Elderly foreigners** | 41–50% | 24 | 20.2 |
| **as a % of the elderly** | +50% | 47 | 39.5 |
| | Total | 119 | 100.0 |

Source: Own elaboration with Local Register data.

4.  Our data provide some confirmation not only for the spatial concentration of retirement migration but also for a tendency to cluster by nationality. In Figure 2 we can see that there are 'British spots' and 'German spots', and although this is by no means

a radical separation. It is interesting to note that places where more than 60% of the foreign residents are from the UK tend to have very little German residents (less than 10%) and vice versa, places where German residents are substantial (more than 30%) tend to have less UK residents.

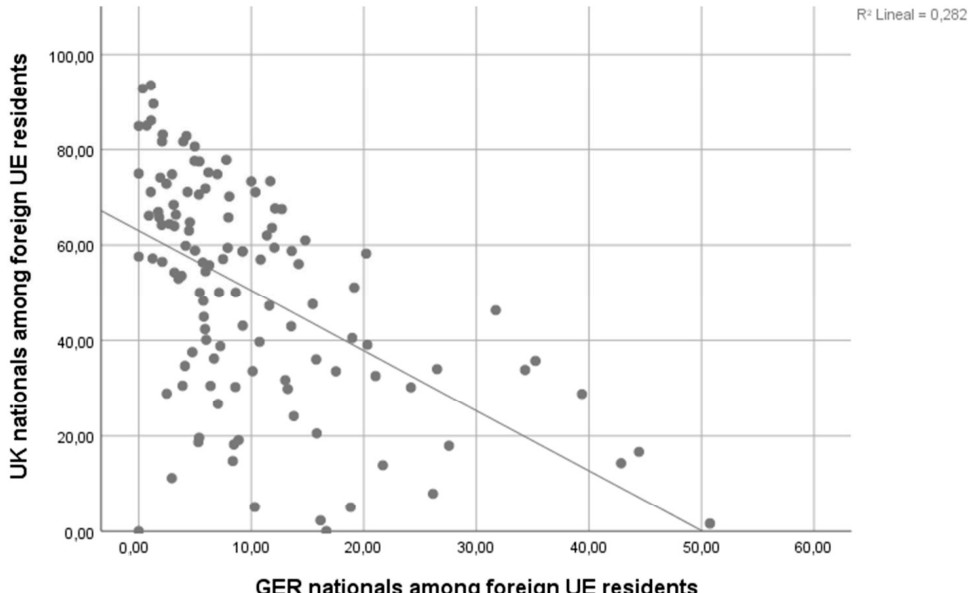

**Figure 2.** Spatial concentration by nationality. Each dot is a municipality. In the horizontal axis one can see the share of German nationals among all foreign residents of 65 years old and more. In the vertical axis one can see the share of UK nationals among all foreign residents of 65 years old and more. The line is the regression line for a linear regression model with the two variables. Source: Own elaboration with Local Register data.

*4.2. European Elderly as Users of Local Social Services*

Notwithstanding the high number of elderly foreigners, in Table 2 we can see that, in most places, they are not an important group among the claimants of Social Services and benefits. Table 2 is based on the information given to us by qualitative phone interviews with coordinators of local Social Services covering 80 of the 119 European Retirement Places. Local Social Services do not register the nationality of claimants when they are citizens of an EU country, and hence the coordinators gave us the share of elderly Europeans among claimants as an approximation based on the previous years' experience. However, the information given helps to evaluate the importance of retirement migrants among users of Social Services, and the first conclusion is that this importance varies a lot across municipalities.

**Table 2.** Retirement migrants among users of local Social Services.

| | | **Number of Municipalities** | **Percentage** |
|---|---|---|---|
| | 0–10% | 43 | 53.8 |
| **European** | 11–20% | 24 | 30.0 |
| **elderly as % of Social** | 21–30% | 6 | 7.5 |
| **Services claimants** | More than 30% | 7 | 8.8 |
| | Total | 80 | 100.0 |

Source: Primary data from our telephonic interviews. We have data only for 80 out of 119 municipalities. In some municipalities, the coordinator of Social Services could not provide any approximation of the number (or share, or percentage) of elderly foreigners among the claimants. In other cases, they did not want to be interviewed.

We are studying only places with very high numbers of retirement migrants, and the fact that in half of the municipalities European retirees rarely contact the Social Services is noticeable. Of course, there are municipalities in which claims from this group amount to

almost one third of all claimants, but these are rare cases (less than 10% of the European Retirement Places).

In addition to quantifying the share of European elderly among claimants, in the telephonic interviews with coordinators of Social Services, we asked them to explain more details of their relationship with this group of residents. From these open-ended questions, we know that most Social Services' coordinators consider that retirement migrants do not approach Social Services as much as locals do. In no interview have we found comments on over-use of local Social Services, and none of the interviewees appear to have the idea of 'welfare tourism' in mind. The risk for the SAAD (public program for dependency support) financial capacity of having to deal with retirement migrants has been completely absent in the phone interviews made with the coordinators of Social Services. In general, they think that most of the older Europeans living in Spain have better social protection from their native countries and enjoy higher pensions than locals, which explains why they do not appear much by Social Services. Although many coordinators of Social Services point out that they do not know much about retirement migrants, they believe that this group of residents are not particularly in need of help. They believe that they use private companies run by people who speak their own language, and that they have dense support networks. Coherent with previous studies, social workers perceive retirement migrants as hermetic communities that stand out for being very supportive with each other.

> *It is that they hardly come because they are self-sufficient, they aren´t demanding Social Services, they protect each other. They usually help each other, between friends. ( . . . ) They aren´t a group that usually request aid, we´ve had a case, but just once.*

Municipality in the province of Alicante (Region of Andalucía).

In line with previous research, coordinators of Social Services describe European retirees as people who are not interested in integrating in the municipality where they reside, who do not learn Spanish despite having been in Spain for a long time, and who live in "bubble communities": united communities of people with the same age and nationality in which networks of mutual support are strong. For the most part, they only go to the Social Services Center to carry out specific procedures such as the "+65 discount card" or to request information. An elderly European that approaches local Social Services looking for economic support is a rare case.

> *If they have money, they hire people from their community, well-known professionals. If they don´t have financial resources, then they come to Social Services and request the service of "home help" for cleaning, etc. And also [they demand help with] Dependency [claiming care support to the SAAD program in case of dependency] and Registration procedures.*

Municipality in the province of Tenerife (Region of Canary Islands).

> *( . . . ) We learn of the existence of these people when they claim dependency benefits. They become disabled, they find themselves alone, and we learn about their situation by a neighbor or by someone else that contacts us. Before that, they never approach the Social Services.*

Municipality in the province of Málaga (Region of Andalucía).

When we asked about the nature of the claims made by the, generally few, retirement migrants that contact the Social Services, the answers always mention the SAAD, the public program which grants support in cases of dependency.

> *Q. Is there a common pattern regarding the needs of elderly Europeans?*

> *What we do most is Information Service. We process the "Andalucía 65 card" [discount card for those over 65]. Of every 10 cards, half are for English people. And especially issues related to Dependency. Of the 200 new applications [this year], 8–10% come from this group. What we do most is Dependency. They lose autonomy, and then is when they really come to us. Disability and everything related to technical aids.*

Municipality in the province of Almería (Region of Andalucía).

*Q. The older Europeans you have cared for, what kind of needs did they have?*

*Dependency problems, need for Home Assistance . . . They are alone; they do not have a family. They arrived [to Spain] when they were young, they were renting their homes . . . but they have grown older, with alcoholism problems, they are dependent people, they have run out of money . . . with non-contributory pensions. Alcohol [1], physical decline . . . They need help because they are uprooted.*

Municipality in the province of Málaga (Region of Andalucía).

However, to be able to access services and benefits under the SAAD, one needs to have an evaluation of the level of dependency as well as an economic evaluation. This process is carried out by social workers and requires the claimant to provide several documents, which is mentioned as one of the major problems dealing with retirement migrants.

*Q. What is the most difficult thing to manage in these cases?*

*A. With the language barrier, the most complicated thing is to manage the Dependency application, because you have to ask for more documentation, medical reports, . . . the rest of things do not present problems, it is pure bureaucratic management, they [elderly Europeans] make easy requests.*

Municipality in the province of Almería (Region of Andalucía).

*4.3. Empirical Evaluation of Hypotheses*

Our first hypothesis stated that levels of registration in Spain by EU elderly should correlate with levels of claims to the local Social Services by this population. Registration would then be employed as a means to use the destination country's welfare system.

In Figure 3 and in Table 3, we can see quite a large variation in the impact that retirement migrants are having in local Social Services. However, there is no relationship with the share of the elderly population that holds a non-Spanish passport, i.e., having huge numbers of foreign elderly registered as residents does not equal having many European retirees claiming social assistance

Since there is almost no relationship between the share of elderly foreigners in a municipality and the number of elderly foreigners that make claims to local Social Services, registration does not seem to be solely a strategy to use the Spanish Welfare State.

Our second hypothesis stated that UK elderly citizens living in Spain use Spanish Social Services relatively more than German elderly citizens living in Spain. According to the phone interviews with coordinators of Social Services, in almost all municipalities, UK retirement migrants were the main group of elderly Europeans using the Social Services. This does not mean that the UK elderly massively approach local Social Services. We already said that under-use is the pattern across all municipalities. However, even if only a few do use local Social Services, the British are always among them. This is not surprising, because UK citizens are the largest group of elderly foreigners in Spain (Figure 1), and its superior presence among users of Social Services can just reflect its higher number. To evaluate if UK citizens tend to use local Social Services more than German citizens, we need to see if municipalities with a high share of UK citizens among its residents are characterized by a high presence of UK seniors among claimants of Social Services.

If UK nationals strategically chose Spanish Social Services, whilst Germans strategically chose not to use them (either because they prefer not to register and keep access to German social rights or because they can transfer some help for dependency from Germany), we shall find a relationship between the share of British among foreign elderly and the share of retirement migrants among users of the Social Services.

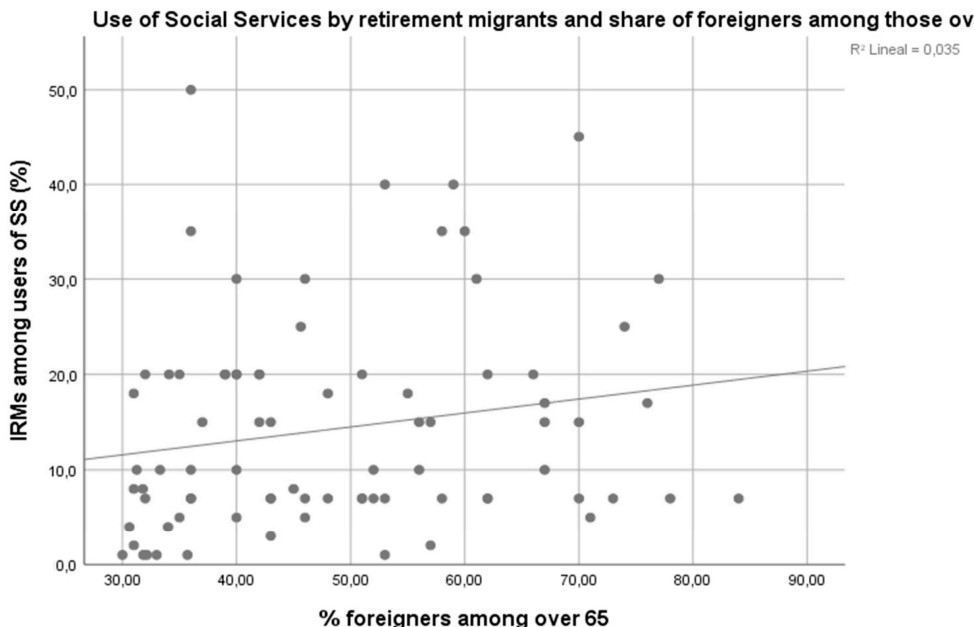

**Figure 3.** Share of foreigners among the elderly and share of retirement migrants among users of local Social Services. Source: Own elaboration with primary data from our telephonic survey with Social Services' coordinators and data from Local Registers. Each dot is a municipality. In the horizontal axis one can see the share of foreigners among elderly residents in each municipality; In the vertical axis one can see the share of European elderly (non-Spaniards) among claimants of Sociale Services. The line is the regression line for a linear regression model with the two variables.

**Table 3.** European elderly among user of Social Services and share of foreigners in the municipality (% of local elderly).

| | | Elderly Foreigners in the Municipality (% of Local Elderly) | | | Total |
|---|---|---|---|---|---|
| | | **30–40%** | **41–50%** | **+50%** | |
| **European elderly as % of Social Services claimants** | 0–10% | 19 61.3% | 7 50.0% | 17 48.6% | 43 53.8% |
| | 11–20% | 9 29.0% | 5 35.7% | 10 28.6% | 24 30.0% |
| | 21–30% | 1 3.2% | 2 14.3% | 3 8.6% | 6 7.5% |
| | More than 30% | 2 6.5% | 0 0.0% | 5 14.3% | 7 8.8% |
| Total | | 31 100.0% | 14 100.0% | 35 100.0% | 80 100.0% |

**Source:** Own elaboration with primary data from our telephonic survey with Social Services' coordinators and data from Local Registers.

However, Figure 4 shows that there is no relation between the share of UK citizens registered as residents and the share of retirement migrants claiming social benefits in Spain. We have municipalities where more than 80% of the foreign residents are from the UK and that show very low rates of retirement migrant claims.

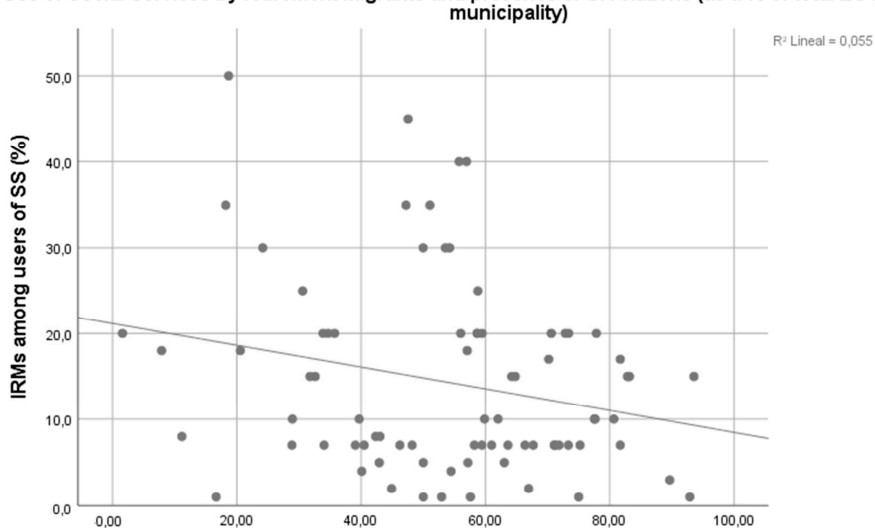

**Figure 4.** Retirement migrants among users of Social Services and share of UK elderly citizens among the elderly. Source: Own elaboration with primary data from our telephonic survey with Social Services' coordinators and data from Local Registers. Each dot is a municipality. In the horizontal axis one can see the share of UK nationals among foreign residents. In the vertical axis one can see the share of European elderly among claimants of the Social Services. The line is the regression line for a linear regression model with the two variables.

**Hypothesis 3.** *UK citizens are more present among users of local Social Services even in municipalities that are key destination spots for Germans.*

According to our database, in municipalities where German citizens are more than 15% of the registered elderly foreigners, they are also present among users of Social Services (see Figure 5). In municipalities where they are more than 30% of elderly foreign residents, they are the main group of retirement migrants using the Social Services. The hypothesis is hence not confirmed. Germans do register and do use Spanish Social Services. However, similar to the English case, the use that German retirees make of public Social Services varies a lot across municipalities and it is not related to the share of Germans among the foreign elderly registered as residents.

*4.4. Alternative Explanation for the Differential Impact of Retirement Migration on Local Social Services*

Up till now, we have seen that there is no relationship between the share of the elderly population that come from a foreign country and the use that retirement migrants make of public Social Services. There are municipalities with huge numbers of retirement migrants and virtually no presence of them in the local Social Services, and municipalities where retirement migrants are not a large group among the elderly but where they do frequently approach the Social Services.

We have also seen that British elderly are, in most places, the largest group of retirement migrants using Social Services, but when Germans are a large group in a municipality, they are also present among those who apply for public help. Henceforth, we do not find support for the idea of retirement migrants as welfare maximizers that only register in a municipality when they plan to use the local public services.

The variation in the impact that retirement migrants have on local Social Services remains to be explained. We can put forward a tentative explanation on the 'classical' lines of social class: there is a correlation between the average income of the municipality and the number of retirement migrants that approach the Social Services (low-income

municipalities tend to have a higher impact of retirement migrants on the Social Services). This relationship can be appreciated in Figure 6, although it is quite blurry.

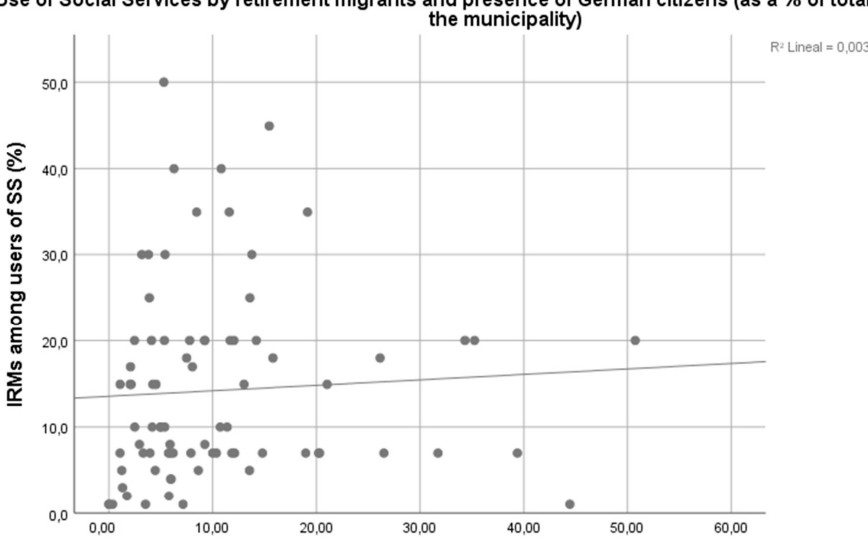

**Figure 5.** Retirement migrants among users of Social Services and share of German elderly citizens. Source: Own elaboration with primary data from our telephonic survey with Social Services' coordinators and data from Local Registers. Each dot is a municipality. In the horizontal axis one can see the share of German nationals among foreign residents. In the vertical axis one can see the share of European elderly among claimants of the Social Services. The line is the regression line for a linear regression model with the two variables.

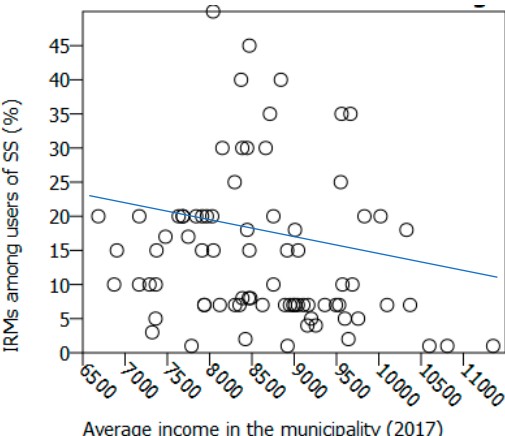

**Figure 6.** Use of Social Services by retirement migrants and average income of the municipality. Each dot is a municipality. The line is the regression line for a linear regression model with the two variables.

The picture of retirement migrants as 'welfare maximizers' is not coherent with this finding. Retirement migrants, as Spanish migrants, need help from the public Social Services more often when they are of low income. Moreover, we can imagine that European retirees with low pensions chose to settle in places where prices are not very high (e.g., housing prices).

One could interpret this result as the existence of 'cheap' retirement places and 'rich retirement places'. Low-income European pensioners chose low-income Spanish municipalities because prices (housing, food, catering sector) are more affordable, both for the local residents and for them. In these places, European retirees may have, on average, low pensions (as compared to average European retirees) and will have to rely on public

services when they lose autonomy. By contrast, expensive municipalities attract IRMs with a higher purchasing power that, when in need, may turn to the private market for help.

## 5. Discussion and Conclusions

Our research has implications for academic studies on retirement migration, for practitioners and for policy makers.

Regarding academic studies, we show that many retirement migrants stay in Spain even after losing some autonomy and with various degrees of dependency. Autonomy problems do not push retirement migrants back to their native countries, and hence there is another story about retirement migrants that is more problematic than the previous literature indicates. Authors such as Kelly Hall already pointed to this phenomenon. Our own fieldwork accords with her findings. A new path opens up in the field of international retirement migration, and further research may be necessary to untangle the complex issues of elderly care in a foreign country for those who decide to stay.

For practitioners of the Social Services, our research is helpful because it provides key information about the needs of retirement migrants and about the situations in which they may require social assistance. Importantly, what we have seen is an extreme under-use of local Social Services by retirement migrants. There is no 'welfare tourism' at all. Since 2007, Spain put into place a universal public program to guarantee care for all the elderly registered in Spain (SAAD). However, retirement migrants apply for this program less frequently. In fact, if they do access the program, it is due to emergency and critical situations that require immediate assistance, but rarely on a voluntary basis. Going back to the title of our paper, what we found is misinformation about social benefits rather than maximization of social rights.

Policy makers could extract two main conclusions from our work: First, retirement migrants are a large group of residents in Spain and many want to stay even after dependency problems set in; secondly, information is the main challenge. Language makes things more complicated when dealing with social workers but the real problem is that retirement migrants seem to be unaware of the existence of the Spanish system of Social Services; that they have expectations based on their own systems and cultures, and that normally, they are not well prepared to claim social rights, i.e., they do not qualify for benefits because they are not registered; they have not performed the bureaucratic necessary steps to comply with the criteria of the programs, etc. This lack of information results in a high rate of retirement migrants that lack the necessary resources to respond to the challenges of growing old in a foreign country and lead a dignified life. Hence, information redounds to the benefit of all.

**Author Contributions:** Conceptualization, I.C.; methodology, I.C.; software, I.C. and R.M.-C.; formal analysis, I.C., V.P. and R.M.-C.; investigation, I.C., V.P. and R.M.-C.; data curation, A.H.; writing—original draft preparation, I.C.; writing—review and editing, I.C. and V.P.; project administration, I.C. and V.P.; funding acquisition, I.C. All authors have read and agreed to the published version of the manuscript.

**Funding:** This research was funded by the Spanish Ministry of Science, under the National Framework for R&D. Grant number: RTI2018-098003-A-I00.

**Institutional Review Board Statement:** The study was conducted in accordance with the Declaration of Helsinki. It was funded by the Spanish Ministry of Science and the evaluation process includes an ethical assessment.

**Informed Consent Statement:** All subjects involved in the study were informed about the nature of the research and about the public funding granted to the project. All interviewees explicitly agreed to be interviewed and recorded. All were informed that their information would always remain anonymous and that they could withdraw from the research at any time.

**Data Availability Statement:** Data not yet available for public use.

**Conflicts of Interest:** The authors declare no conflict of interest.

## Notes

1      The issue of some retirement migrants having alcohol problems has been mentioned in other interviews and is also present in previous studies but we would need much more research to actually evaluate the extent of the problem. Stereotypes about Germans, English, and Nordic people as heavy drinkers may be at play. Some previous studies mentioning the issue: [17,18,51].

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
