# Peer review of "The Best Welfare Deal: Retirement Migrants as Welfare Maximizers"

_societies, doi:10.3390/soc13040102_

Round 1
Reviewer 1 Report
This article is of great interest and deals with a neglected aspect of International Retired Migrants as it is their social issues. There are some references within the text that do not appear later in the reference list. There are also some phrases with a diferent font size from the rest of the document. It would be great if map 1 had a better resolution and a source. Graph 6 has no number, plus the X axis is in Spanish. In the line 654 there appear "graphs X and X" which need to be controlled. Graphs in lines 661 and 662 are not numbered and the Y axis is in Spanish. Conclusions could be a bit more extensive.
Author Response
R1
This article is of great interest and deals with a neglected aspect of International Retired Migrants as it is their social issues.
There are some references within the text that do not appear later in the reference list. There are also some phrases with a different font size from the rest of the document. It would be great if map 1 had a better resolution and a source. Graph 6 has no number, plus the X axis is in Spanish. In the line 654 there appear "graphs X and X" which need to be controlled. Graphs in lines 661 and 662 are not numbered and the Y axis is in Spanish. Conclusions could be a bit more extensive.
Dear reviewer, thank you very much for your positive evaluation of our paper. We apologize for the mistakes in formal aspects such as graph numeration, font size, etc. We have reviewed the paper trying to fix all editing problems:
-we have checked spelling and grammar.
-we have checked the connection between text and Bibliography and made sure that all references in the text appear also in the Bibliography.
-we have improved the information on the graphs, we have included graph numbers when missing.
-We have translated the X axis of graph 6 to English.
-We have eliminated map 1, since it gave a basic geographical information.
-We eliminated graphs 7 and 8, since they offered redundant information and there are already too many graphs in the text.
-We have expanded the conclusions and included some new reflections.
Reviewer 2 Report
This paper makes a significant contribution to the literature on International Retirement Migration. The focus on EU citizens is beneficial for understanding how retirees find, access, and maintain care in their adopted home nation. Most of my suggestions center on methods and some typos/points of clarification in the paper.
1) Can the authors explain the nature of the online surveys better? You say on line 487 that Table 2 is based on information from coordinators of Social Services at the local level. Were these phone interviews or surveys? The Total given in Table 2 for Number of Municipalities is 80, which you explain, but beneath that you have 727. What is this number referring to? It isn't clear throughout when data is coming from online surveys or from phone interviews or from case studies.
2) Explain the "case studies" more clearly. It is unclear where, when, how these "case studies" were completed and how they use (or differ from) data collected in the surveys or phone interviews.
3) Again, when the very informative and important qualitative data is presented (lines 548-574, for example) it isn't clear whether this was qualitative information people offered in the surveys or whether this comes exclusively from phone interviews. I was left wondering whether the respondents keep regular, comprehensive data of the kind you asked about in this section, or if this qualitative data was more anecdotal and based on their personal/professional experience.
4) One question that remains is whether you can hypothesize about WHY people chose to relocate in their retirement and how this might correlate with their use of local public services. You suggest on line 21 that most retirement migrants do not move for economic reasons, but if your research shows that for some who had a low(er) income or pension in their country of origin are choosing to move then we might assume that this IS an economic reason (see Gambold 2013; Hayes 2014, 2015). "Lifestyle" as a driving force cannot easily be separated from "economic reasons".
5) Another major question is how Brexit has impacted retirees from the UK? Even if this was not a question you asked, I feel it needs to be mentioned.
The following are smaller points of clarification or fixing wording/spelling:
1) line 198 capitalize Spanish
2) line 357 point c) is unclear and needs to be rewritten
3) line 369 should read "involves much more extensive"
4) line 446 "on the coast"
5) lines 477 - 481, capitalize British, German...
6) line 423 change "for" to "from"
7) line 499 "they did not want" instead of "wanted"
8) Be consistent with Social Workers capitalization (lines 501, 507...)
9) line 513 add apostrophe to "don't"
10) line 514, remove extra space after IRMs
11) line 523 "help for", remove "por"
12) line 529 "Made up of people..." is an incomplete sentence and should be part of the previous sentence.
13) line 543, don't understand having "SS.SS."
14) line 560, is it possible to mention these "alcoholism problems" somewhere? This might be surprising to some readers and deserves a point of clarification and maybe reference to other literature.
15) line 633, "There higher the share" needs to be rewritten.
16) line 674, "what we have seen IS and extreme..."
17) last paragraph in the conclusions could be expanded a little to perhaps suggest that since information is the main challenge, local service providers/governments need to do better at creating and providing multi-language resources for foreign-born retirees.
18) Note #1 needs to be translated to English.
Overall, I really enjoyed reading this paper and think it will be of use to many who are interested in IRM in the EU and elsewhere.
Author Response
R2
We are extremely grateful to Reviewer two for her/his careful reading of our work and for her/his valuable comments and suggestions. As we detail below comment by comment, we have included almost all her/his suggestions in the text.
This paper makes a significant contribution to the literature on International Retirement Migration. The focus on EU citizens is beneficial for understanding how retirees find, access, and maintain care in their adopted home nation. Most of my suggestions center on methods and some typos/points of clarification in the paper.
- Can the authors explain the nature of the online surveys better? You say on line 487 that Table 2 is based on information from coordinators of Social Services at the local level. Were these phone interviews or surveys? The Total given in Table 2 for Number of Municipalities is 80, which you explain, but beneath that you have 727. What is this number referring to? It isn't clear throughout when data is coming from online surveys or from phone interviews or from case studies.
Thank you very much for this suggestion. Indeed, the paper lacked clarity regarding the details of the methods to gather information. Actually, most of the paper is based on results from qualitative phone interviews with coordinators of Social Services. The results from our online survey were not included in this paper. We are very sorry for the confusing part about methods. We have done an effort to re-write the Methodology and also some paragraphs in the Results section so that the nature of our data is clearer. The number 727 from Table 2 is just a typo and has been deleted.
- Explain the "case studies" more clearly. It is unclear where, when, how these "case studies" were completed and how they use (or differ from) data collected in the surveys or phone interviews.
We have done so in the Methods section. Each of the Methods used during the research has been explained with more detail.
- Again, when the very informative and important qualitative data is presented (lines 548-574, for example) it isn't clear whether this was qualitative information people offered in the surveys or whether this comes exclusively from phone interviews. I was left wondering whether the respondents keep regular, comprehensive data of the kind you asked about in this section, or if this qualitative data was more anecdotal and based on their personal/professional experience.
You are totally right. The qualitative information appears suddenly and with no contextualization. We have included information on the origin of the qualitative material present in the paper. The Social Services don’t keep comprehensive data on the nationality of claimants when they are from EU countries, and hence our data is based on their personal experience and the “share of European elderly among claimants” is asked as an approximation based on last year claims.
4) One question that remains is whether you can hypothesize about WHY people chose to relocate in their retirement and how this might correlate with their use of local public services. You suggest on line 21 that most retirement migrants do not move for economic reasons, but if your research shows that for some who had a low(er) income or pension in their country of origin are choosing to move then we might assume that this IS an economic reason (see Gambold 2013; Hayes 2014, 2015). "Lifestyle" as a driving force cannot easily be separated from "economic reasons".
Yes, this is a very interesting comment, and of course it is difficult to disentangle economic reasons and other reasons to move. In our study we found that some retirement migrants are not particularly well-off, but we could not infer from their economic situation that their major driver to move was economic. Actually, in the qualitative interviews from ethnographic case studies the economic motivation has never been mentioned as the main driver for migration.
In any case, we prefer to leave the issue aside since we don’t have enough information to question this idea. We only mention that previous literature defines retirement migration as a movement in which economic reasons are not central, and we have added: (although changes in pension systems can increase the importance of economic drivers)
Another major question is how Brexit has impacted retirees from the UK? Even if this was not a question you asked, I feel it needs to be mentioned.
Indeed, the impact of Brexit on retirement migrants is a big topic, but we leaved it out of this article because we felt that: a) the article was already too long and with too much information; b) the Brexit impact requires a complete article. We are now working in a new article about Brexit and IRM in Spain.
The following are smaller points of clarification or fixing wording/spelling:
1) line 198 capitalize Spanish - DONE
2) line 357 point c) is unclear and needs to be rewritten - DONE
3) line 369 should read "involves much more extensive" - DONE
4) line 446 "on the coast" - DONE
5) lines 477 - 481, capitalize British, German... - DONE
6) line 423 change "for" to "from" - DONE
7) line 499 "they did not want" instead of "wanted"- DONE
8) Be consistent with Social Workers capitalization (lines 501, 507...) DONE
9) line 513 add apostrophe to "don't" – DONE (PARAGRAPH HAS BEEN RE-WRITEN)
10) line 514, remove extra space after IRMs – DONE (PARAGRAPH HAS BEEN RE-WRITEN)
11) line 523 "help for", remove "por" – DONE (PARAGRAPH HAS BEEN RE-WRITEN)
12) line 529 "Made up of people..." is an incomplete sentence and should be part of the previous sentence. – DONE (PARAGRAPH HAS BEEN RE-WRITEN)
13) line 543, don't understand having "SS.SS." - DONE
14) line 560, is it possible to mention these "alcoholism problems" somewhere? This might be surprising to some readers and deserves a point of clarification and maybe reference to other literature.
We have included a footnote and the following references:
Huete, R., Mantecon, A. and Estevez, J. (2012). Challenges in lifestyle migration research: reflections and findings about the Spanish crisis. Mobilities, 8, 3, 331–48.
Hall, K. y Hardill, I. (2016). Retirement migration, the 'other' story: caring for frail elderly British citizens in Spain. Ageing and Society, 36(3), 562-585
Gavanas, A. (2017). Swedish Retirement Migrant Communities in Spain: Privatization, informalization and moral economy filling transnational care gaps, Nordic Journal of Migration Research, 7(3), 165-171.
15) line 633, "There higher the share" needs to be rewritten. - DONE
16) line 674, "what we have seen IS and extreme..." - DONE
17) last paragraph in the conclusions could be expanded a little to perhaps suggest that since information is the main challenge, local service providers/governments need to do better at creating and providing multi-language resources for foreign-born retirees.
We expanded the conclusions and, to gain space for it, we merged hypothesis 3 and 4 in only one hypothesis.
18) Note #1 needs to be translated to English. – Note 1 was the original definition (in Spanish) of “European Retirement Place”. The English translation is in the text, and we thought that it was good to include also the original Spanish text. We believe now that this is confusing and we deleted Note 1.
Overall, I really enjoyed reading this paper and think it will be of use to many who are interested in IRM in the EU and elsewhere.